# Enhancing efficiency of protein language models with minimal wet-lab data through few-shot learning

Ziyi Zhou[1,2,7], Liang Zhang[1,7], Yuanxi Yu [1,7], Banghao Wu[3,7], Mingchen Li [4,5], Liang Hong [1,2,4,6] ✉ & Pan Tan [1,2,4] ✉

Accurately modeling the protein fitness landscapes holds great importance for protein engineering. Pre-trained protein language models have achieved state-of-the-art performance in predicting protein fitness without wet-lab experimental data, but their accuracy and interpretability remain limited. On the other hand, traditional supervised deep learning models require abundant labeled training examples for performance improvements, posing a practical barrier. In this work, we introduce FSFP, a training strategy that can effectively optimize protein language models under extreme data scarcity for fitness prediction. By combining meta-transfer learning, learning to rank, and parameter-efficient fine-tuning, FSFP can significantly boost the performance of various protein language models using merely tens of labeled single-site mutants from the target protein. In silico benchmarks across 87 deep mutational scanning datasets demonstrate FSFP's superiority over both unsupervised and supervised baselines. Furthermore, we successfully apply FSFP to engineer the Phi29 DNA polymerase through wet-lab experiments, achieving a 25% increase in the positive rate. These results underscore the potential of our approach in aiding AI-guided protein engineering.

Proteins play an indispensable role in biological activities. Due to their attributes as biocatalysts, which are green, efficient, and cost-effective, the demand for their applications in scientific research and industrial production is steadily increasing[1–4]. However, most wild-type proteins, directly obtained from the biological species, cannot be directly applied in industrial conditions, as some of their physicochemical properties, such as stability, activity, and substrate specificity, are not good enough. Protein engineering seeks to excavate proteins with properties useful for specific applications. Traditional protein engineering, relying on methods like directed evolution and rational design, seeks to enhance these properties[5,6]. Directed evolution,

although powerful, faces challenges in screening vast mutant libraries due to high-throughput assay constraints in terms of setup complexity and costs[7,8]. Rational design, despite its reduced experimental requirements, is often limited by the unavailability of detailed structural knowledge and mechanistic insights[9,10]. In recent years, deep learning has shown great potential in uncovering the implicit relationships between protein sequences and their functionality, i.e., fitness, thus being helpful to efficiently explore the vast design space.

Generally, deep learning approaches can be categorized into supervised and unsupervised models, with the main distinction being whether the training data require manually collected labels[11–13].

[1]School of Physics and Astronomy, Shanghai Jiao Tong University, Shanghai 200240, China. [2]Shanghai National Center for Applied Mathematics (SJTU Center) & Institute of Natural Sciences, Shanghai Jiao Tong University, Shanghai 200240, China. [3]School of Life Sciences and Biotechnology, Shanghai Jiao Tong University, Shanghai 200240, China. [4]Shanghai Artificial Intelligence Laboratory, Shanghai 200232, China. [5]School of Information Science and Engineering, East China University of Science and Technology, Shanghai 200237, China. [6]Zhang Jiang Institute for Advanced Study, Shanghai Jiao Tong University, Shanghai 201203, China. [7]These authors contributed equally: Ziyi Zhou, Liang Zhang, Yuanxi Yu, Banghao Wu. ✉e-mail: hongl3liang@sjtu.edu.cn; tpan1039@alumni.sjtu.edu.cn

Pre-trained protein language models (PLMs) are the most trending unsupervised approaches to fitness prediction. These models, such as ESM-2[13], ProGen[14], SaProt[15], and ProtT5[16], trained on the expansive protein universe, can estimate probability distributions for various protein sequences independent of experimental data. This capability facilitates the prediction of mutational effects but is limited in accuracy. Since these models fundamentally represent statistical characteristics of natural protein sequences found in nature, their zero-shot likelihood scores for mutation fitness essentially measure how similar a mutant protein sequence is to natural proteins or a particular protein family. While this measure can predict certain natural protein properties like solubility and stability[12,17,18], it inherently lacks the capability to predict non-natural catalytic properties, such as the catalysis of non-natural substrates or the production of non-natural products[19].

Supervised deep learning models, in contrast, have recently shown high accuracy in predicting protein fitness[11,20,21]. Based on their strong ability to extract both local and global features of the proteins, they could construct more accurate sequence-fitness correlations by training on sufficient labeled data. However, these models are heavily reliant on extensive data derived from expensive, high-throughput mutagenesis experiments[22,23], posing a significant challenge for most proteins. Recently, Hsu et al.[24] developed an efficient ridge regression model that combines the one-hot features of amino acids and the probability density feature calculated by an unsupervised model. When training on limited labeled data, it demonstrates improved performance against more sophisticated and expensive methods. However, one-hot features are not informative enough to represent the relationships between different residues. Besides, as a linear model, ridge regression might have difficulty in learning complex patterns that affect protein fitness. Therefore, it is meaningful to develop new strategies to effectively fine-tune PLMs with scant wet-lab data in protein engineering, where the advantages of both unsupervised and supervised approaches can be fused.

In this work, we leverage the synergistic methodologies of meta-transfer learning (MTL)[25], learning to rank (LTR)[26,27], and parameter-efficient finetuning[28,29] to develop a versatile approach for training PLMs. Our approach, named FSFP (**F**ew-**S**hot Learning for Protein **Fit**ness **P**rediction), is notable for its reliance on a minimal labeled dataset for the target protein, comprising merely tens of random single-site mutants. With FSFP, this streamlined dataset can substantially enhance the accuracy of the trained model to predict mutational effects. To validate our approach, we conduct in-silico benchmarks using various representative PLMs, including ESM-1v, ESM-2, and SaProt. Although FSFP is theoretically compatible with any PLMs, our selection of models for testing is primarily influenced by practical considerations, particularly computational efficiency and resource constraints. Our methodology demonstrates remarkable performance on ProteinGym[30], a benchmark including an extensive range of 87 deep mutational scanning (DMS) datasets, showing robustness when adapting to different PLMs and proteins. This is achieved in comparison with both unsupervised and supervised models trained on tens of data. In particular, our approach enhances the performance of the PLMs by up to 0.1 on average Spearman correlation with merely 20 labeled single-site mutants from the target protein. Moreover, FSFP is applied to engineer the Phi29 DNA polymerase through wet-lab experiments where both the average melting temperature ($T_m$) and positive rate of the top 20 predictions from ESM-1v are improved. These results underscore its efficiency in data utilization, indicating its potential in aiding AI-guided protein engineering.

## Results

### Transferring PLMs with limited training data via FSFP
FSFP leverages meta-learning to better train PLMs in a label-scarce scenario. Meta-learning aims to train a model that can rapidly adapt to a new task using only a few training examples and iterations, by accumulating experience from learning multiple tasks[25,31,32]. To build the training tasks required for meta-learning, we search for existing labeled mutant datasets that are potentially helpful to predict the variant effects on the target protein, as well as generate pseudo labels through multiple sequence alignment (MSA) for the candidate mutants (Fig. 1a and "Methods"). In this stage, the wild-type sequences or structures of the target protein and the ones in the database are first encoded into embedding vectors by PLMs. ProteinGym is used as the database to retrieve from because it is the largest public collection of DMS datasets at the time of writing. After that, the associated datasets of the top two proteins that are closest to the target protein in the vector space are selected to form the first two tasks. This is motivated by the fact that the fitness landscapes of similar proteins may share similar properties[33]. On the other hand, existing literature shows that it is promising to predict the effect of genetic variation using MSA[11,18,30,34–38]. We thereby use an alignment-based method, GEMME[34], to utilize MSA information of the target protein, and score the candidate mutants of interest to build the dataset of the third task. The labeled data of these tasks are randomly split into training and testing data, respectively. In this way, we expect the meta-trained model to learn to utilize the target training data from both evolutionary information and similar fitness landscapes.

We apply model-agnostic meta-learning (MAML)[32], a popular gradient-based meta-learning method, to meta-train PLMs on the built tasks (Fig. 1b and "Methods"). In effect, MAML learns to find the optimal initial model parameters such that small changes in them will produce large improvements on the target task. The meta-training procedure has two levels of optimization in each iteration, and will eventually turn the PLM into a meta-learner for initialization (Fig. 1b, right). In the inner-level optimization, a temporary base-learner is initialized by the current meta-learner and then updated into a task-specific model using the training data of a sampled task. In the outer-level optimization, the test loss of the task-specific model on that task is used to optimize the meta-learner.

PLMs typically use heavily parameterized Transformer[39] models as the base architecture and pre-train on large-scale unlabeled protein sequences[12–14,16,18,40] or structures[15,41]. However, when finetuning on very few labeled training data, they are likely to suffer from catastrophic overfitting. Therefore, FSFP utilizes low-rank adaptation (LoRA)[28] to inject trainable rank decomposition matrices into PLMs with their original pre-trained parameters frozen, and all of the model updates are constrained to these small number of trainable parameters (Fig. 1b, left and "Methods").

After meta-training, a good initialization of the LoRA parameters is obtained, and we finally transfer the meta-trained PLMs to the target few-shot learning task, i.e., learning to predict the mutational effects of the target protein using its limited labeled data. Unlike conventional approaches to train supervised protein fitness predictors, which formalize this to a regression problem[11,20,21,24], FSFP treats it as a ranking problem and leverages the LTR technique (Fig. 1c and "Methods"). In protein engineering and directed evolution, the most important index is whether a mutation enhances the functional fitness of the existing protein. Therefore, instead of focusing on the precise score values of mutations, their relative effectiveness or ranking order should hold greater significance. Specifically, FSFP learns to rank the fitness by computing ListMLE loss[27], which is defined by the likelihood of the permutation of a correct ranking. In each iteration, the model is trained to fix its predictions for one or more sampled data subsets towards the ground truth permutation. The above training scheme is adopted in both the transfer learning stage (using the target training data) and the inner-level optimization during the meta-training stage (using the training data of the auxiliary tasks).

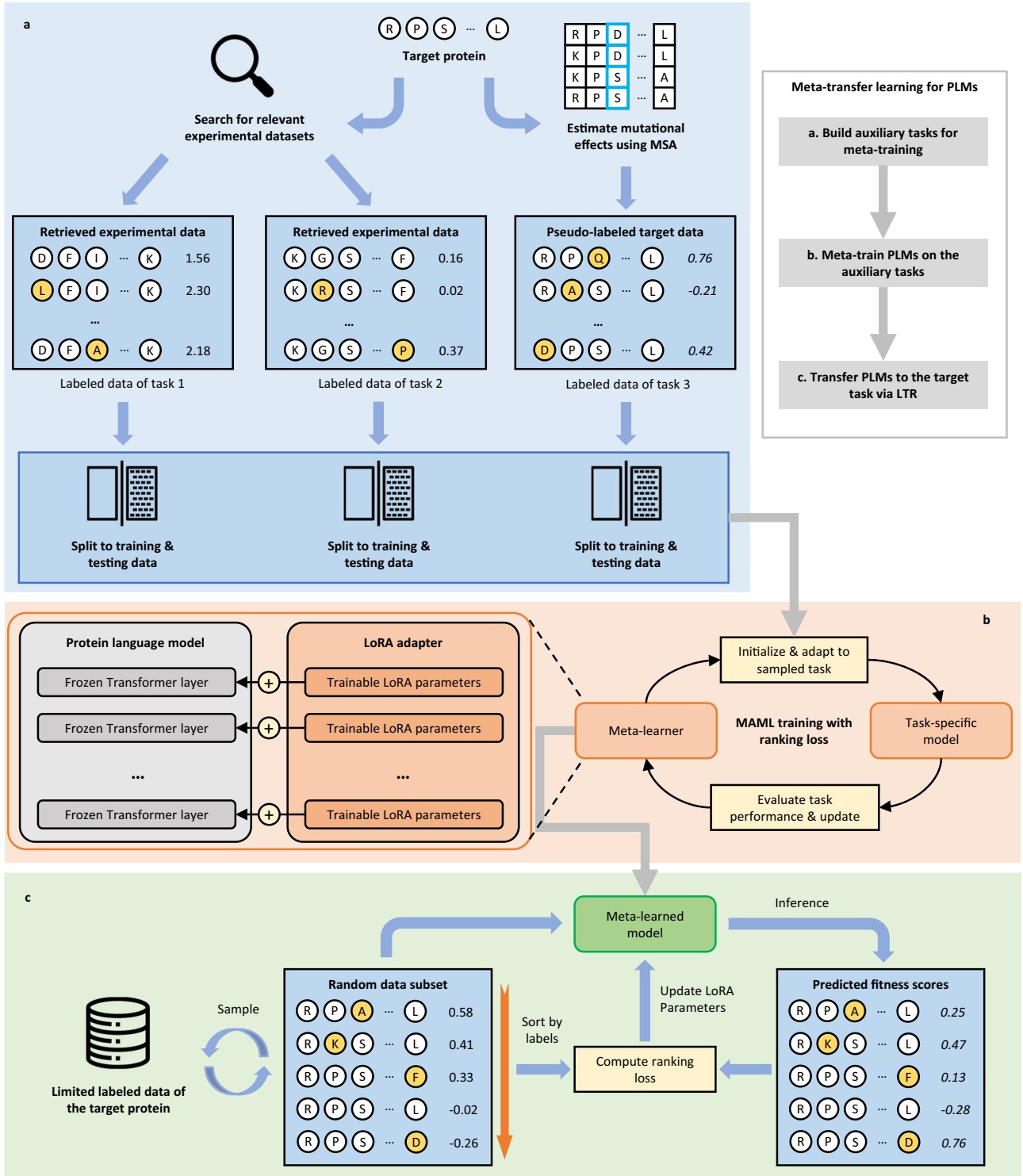

**Fig. 1 | Overview of FSFP.** FSFP includes three stages: building auxiliary tasks for meta-learning, meta-training PLMs on the auxiliary tasks, and transferring PLMs to the target task. **a** Based on the wild-type sequence or structure of the target protein, the labeled mutant datasets of two similar proteins are retrieved to be the first two tasks. In addition, an MSA-based method is used to estimate the variant effects of the candidate mutants as pseudo labels for the third task. **b** MAML algorithm is used to meta-train the PLM on the built tasks and eventually optimizes it into a meta-learner that provides good parameter initialization for the target task (right). To prevent PLMs from overfitting on small training data, LoRA is applied to constrain model updates to a limited number of parameters (left). **c** The meta-trained model is then transferred to the target few-shot learning task. FSFP treats fitness prediction as a ranking problem, and leverages the LTR technique for both transfer learning and meta-training. It trains PLMs to rank the fitness by computing a listwise ranking loss between their predictions and the ground truth permutation.

## Benchmark setup

We evaluate model performance on the substitution benchmark of ProteinGym, which consists of about 1.5 M missense variants from 87 DMS assays. Among these 87 datasets, 11 contain multi-site mutants. ProteinGym is originally used for evaluating the zero-shot performance of PLMs, and we turn it into a few-shot learning benchmark as follows. For each dataset in the benchmark, we first randomly select 20 single-site mutants as an initial training set. Then we expand the training set size to 40 by sampling another 20 single-site mutants. The training sets with sizes of 60, 80, and 100 are built accordingly. For each training set, all the remaining data (or part of them if specified) are used as a test set, and we use cross-validation on the training data to determine the training hyperparameters ("Methods"). The data splitting process is repeated five times with different random seeds, and we average the model performance over different splits of a certain training size. In most experiments, the predictive performance is measured by two metrics: Spearman rank correlation and normalized discounted cumulative gain (NDCG)[42] with the fitness labels as ground truth.

Our main baseline on few-shot protein fitness prediction is the ridge regression approach introduced by Hsu et al.[24]. It uses one-hot encoded site-specific residue features and the fitness score predicted by an existing evolutionary probability density model as the input features to train a ridge regression model. Although it is simple, it turns out to be more effective than other supervised learning approaches in such low-resource scenarios. When applying FSFP to a foundation model for evaluation, we compare it to the ridge regression-augmented version of this model. We use the official implementation of this baseline to ensure its performance.

## All components of FSFP contribute positively to few-shot learning

To thoroughly evaluate the impact of different components that make up FSFP, we conduct an ablation study taking ESM-2 as the foundation model. The model size is chosen to be 650 M, while other sizes are also evaluated and FSFP keeps achieving better performance on larger models (Supplementary Fig. 1a). In detail, we compare FSFP with the following training strategies:

(1)  LTR + LoRA + MTL (no MSA) is a variant of FSFP that does not depend on MSA to build auxiliary tasks. It replaces the third task of FSFP with another labeled dataset retrieved from the database.
(2)  LTR + LoRA is a variant of FSFP that transfers the LoRA-adapted model to the target task via LTR without meta-training on auxiliary tasks.
(3)  MSE denotes fine-tuning the entire PLM with fitness labels as done by Rives et al.[40], which uses the log-likelihood difference between the mutant and the wild type as a predictor.
(4)  MSE + LoRA further enhances the MSE method with LoRA.
(5)  Ridge regression denotes the approach proposed by Hsu et al.[24].
(6)  We also perform zero-shot inference using the PLM following Meier et al.[12].

Compared with zero-shot inference, MSE exhibits major degradation in the average predictive performance (Fig. 2a and Supplementary Fig. 1b, c). This can be attributed to the quick overfitting caused by directly fine-tuning the whole model on small training datasets (Supplementary Fig. 2b, c). Equipping LoRA significantly mitigates such negative impact overall due to its efficient parameterization for minimal adjustments. However, its performance still fails to match the original unsupervised model until the training set

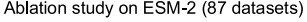

Ablation study on ESM-2 (87 datasets)

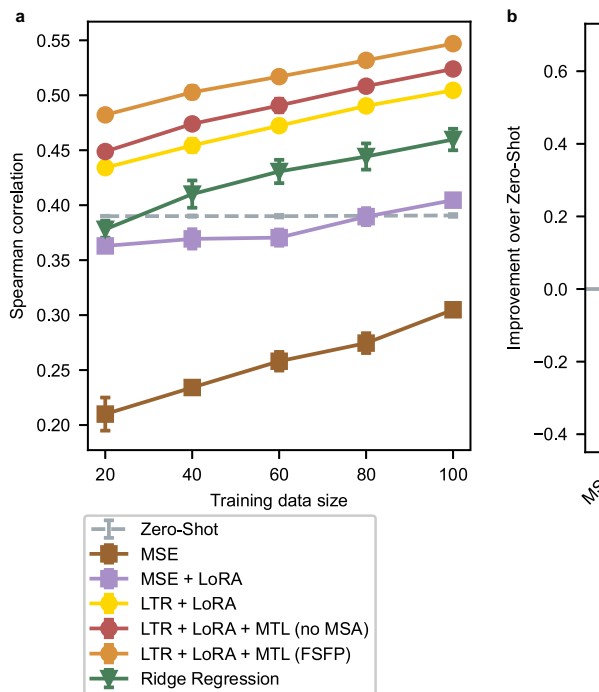

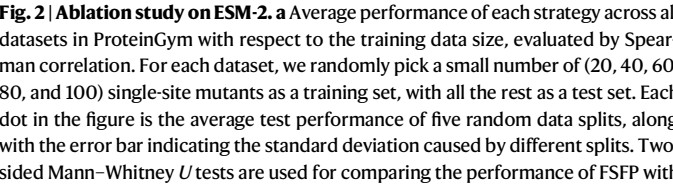

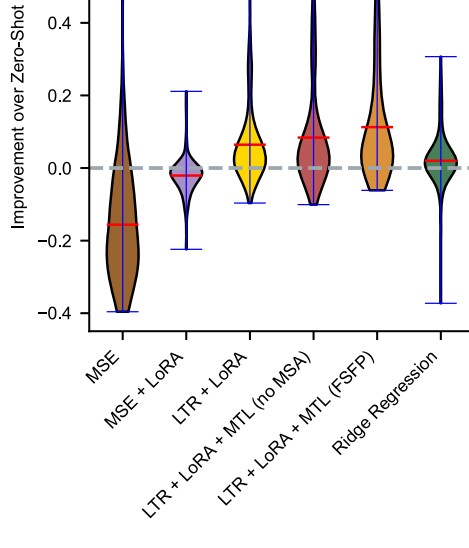

**Fig. 2 | Ablation study on ESM-2. a** Average performance of each strategy across all datasets in ProteinGym with respect to the training data size, evaluated by Spearman correlation. For each dataset, we randomly pick a small number of (20, 40, 60, 80, and 100) single-site mutants as a training set, with all the rest as a test set. Each dot in the figure is the average test performance of five random data splits, along with the error bar indicating the standard deviation caused by different splits. Two-sided Mann–Whitney *U* tests are used for comparing the performance of FSFP with all other strategies, and the largest *P* value among all training sizes is 0.0079. Analogous results measured by NDCG, Pearson correlation, and MAE are shown in Supplementary Fig. 1b–d. **b** Distribution of the performance improvement in Spearman correlation over zero-shot prediction across all datasets in ProteinGym, with a training size of 40. The performance gain of each dataset is averaged among the five random splits. Source data are provided as a Source Data file.

size reaches 80. As our main baseline, the performance of ridge regression exceeds zero-shot inference across most of the training sizes. This is consistent with the comparison results presented by Hsu et al.[24]. LTR + LoRA outperforms the above methods on all training sizes (Fig. 2a and Supplementary Fig. 1b, c). Meanwhile, on most datasets, LTR greatly boosts the model performance compared with regression (Fig. 2b). As mentioned before, the ranking order of different mutants is often more important than their absolute scores in directed evolutions. On the other hand, in the context of few-shot learning, accurately predicting the exact label values becomes challenging because there is usually a significant difference in their range between training and testing data. In fact, when additionally considering mean absolute error (MAE), even the regression-based methods exhibit poor performance (Supplementary Fig. 1d), rendering their absolute output values impractical for applications. Since the order matters more, the ranking-related metrics are favored for fitness prediction and thus LTR is more suitable.

Using MTL to obtain initial model parameters for few-shot training further increases the performance (Fig. 2a and Supplementary Fig. 1b, c). The improvements of LTR + LoRA + MTL (no MSA) over LTR + LoRA indicate that similar properties in fitness landscapes from other proteins can be helpful. FSFP, which additionally learns the MSA knowledge during meta-training, achieves the best scores on all training data sizes (Fig. 2a and Supplementary Fig. 1b, c). This suggests that apart from the training data of the target protein, the evolutionary information from MSA may also effectively supervise the model in estimating mutational effects. As illustrated by the training curves (Supplementary Fig. 2), with or without MSA, MTL could significantly improve the model performance using very few training iterations compared with other approaches. Besides, the initial model after meta-training can already substantially outperform zero-shot inference without access to the target training data (i.e., with no training iteration) in some cases. These demonstrate that by utilizing FSFP, the model successfully learns useful information from auxiliary tasks and thus can well transfer to the target few-shot learning task.

### FSFP as a general few-shot learning approach for PLMs

FSFP can be applied to any deep learning-based protein fitness predictor that uses gradient descent for optimization, while we focus on PLMs in this work. To validate its versatility, we select three representative PLMs—ESM-1v, ESM-2, and SaProt—as the foundation models to be trained ("Methods"). For each of them, the 650 M version is chosen for evaluation, where the trainable LoRA parameters account for 1.84% of the entire model. We compare FSFP with their zero-shot predictions, as well as the ridge regression approach across all 87 datasets in the benchmark. Since FSFP leverages GEMME to generate pseudo labels for meta-training, we also add GEMME and GEMME augmented by the ridge regression approach as two additional baselines. The test performances on single-site mutants and multi-site mutants are reported separately (Fig. 3 and Supplementary Fig. 3). Note that the best approach reported by Hsu et al.[24] is DeepSequence[36] augmented by ridge regression. However, we find that its overall performance is worse than the ridge regression-augmented GEMME on ProteinGym, so we omit this approach from the figures.

Considering the average performance, the PLMs trained by FSFP consistently outperform other baselines on all training data sizes (Fig. 3a, c). Among them, SaProt (FSFP) emerges as the top performer, while ESM-1v (FSFP) and ESM-2 (FSFP) show comparable performance. Besides, on most datasets in ProteinGym, the best Spearman correlation is achieved by one of the FSFP-trained PLMs (Fig. 3b, d). Compared with zero-shot predictions, FSFP boosts the performance of PLMs on single-site mutants by nearly 0.1 on Spearman correlation using only 20 training examples, and this gap becomes even larger when it comes to multi-site mutants. The improvements keep increasing as the training dataset expands, aligning with the results of the prior ablation

study. This demonstrates the adaptability and effectiveness of FSFP on different foundation models. By contrast, the ridge regression approach fails to noticeably outperform its zero-shot counterparts with 20 training examples (Fig. 3a, c). On multi-site mutants, it shows major negative impacts on the performance of GEMME, ESM-1v, and ESM-2 when the training size is 20. Although its performance also tends to improve as the training size increases, it is consistently behind the FSFP-trained models. Except for the training strategy, this is likely due to the limitation of simple one-hot features and model capacity of the ridge regression approach. It is worth noting that when predicting multi-site mutants, the standard errors of FSFP are much smaller than those of ridge regression, indicating that the former is more stable and reliable when given different few-shot training data.

Notably, the average zero-shot performance of GEMME is better than the PLMs we chose. This is not surprising because PLMs may not always be better than MSA-based methods for predicting variant effects[24,30,38], despite their strong ability to model the statistical characteristics of protein sequences or structures. Therefore, resorting to guidance from a state-of-the-art MSA-based method is reasonable in such a few-shot scenario. As shown in Fig. 3a, the models utilizing FSFP achieve substantial improvements over both GEMME and its ridge regression-augmented version on all training sizes. This suggests that FSFP not only instills the MSA knowledge from GEMME into PLMs, but also successfully combines it with the supervised information from the target training data through MTL. This again verifies the superiority of FSFP as a few-shot learning strategy, especially with exceedingly small training datasets.

### FSFP holds robust generalizability and extrapolation ability

For protein engineering, the effect of mutations whose positions do not occur in existing labeled data is always wondered. Therefore, the capability of the fitness predictors to extrapolate across positions is important for probing mutants with good properties from the enormous sequence space. We examine the extrapolation ability of different approaches by evaluating them on specific subsets of the testing data. In detail, from each original test set, we first select all single-site mutants whose mutated positions are different from those of the training examples, resulting in a more difficult test set of single-site mutants. Then we select the multi-site mutants whose individual mutations have no overlap with the mutations in the training data, resulting in another challenging test set. We do not force all mutated positions of a testing multi-site mutant to be different from the training positions because this will lead to insufficient testing examples for some datasets. Under these settings, we can find that the zero-shot performance of the base models obviously varies with the training set size (Fig. 4). This can be attributed to the greater changes in the test sets: as the training dataset expands, available testing examples satisfying the above conditions may drastically decrease for some datasets, impacting the evaluation results of zero-shot predictions.

When extrapolating to single-site mutants with different positions, the models augmented by ridge regression do not show clear improvements over the base models even with 100 training examples (Fig. 4a). For multi-site mutants, the ridge regression approach fails to effectively enhance GEMME and ESM-2 when the training size is <60, and it exhibits larger standard deviations than FSFP as observed before (Fig. 4c). In stark contrast, the PLMs trained by FSFP continue to score better than all baselines on Spearman correlation across different training sizes. Compared with their zero-shot performance, they show considerable improvements after being trained by FSFP, especially for ESM-1v. Also, the best performer of most datasets is an FSFP-trained model (Fig. 4b, d). These are consistent with the quantitative conclusions illustrated in Fig. 3. Since PLMs have intrinsic strong generalizability owing to their high capacity and embedded knowledge through pre-training, it is possible to boost their performance on hard downstream tasks by proper training paradigms. However, the input

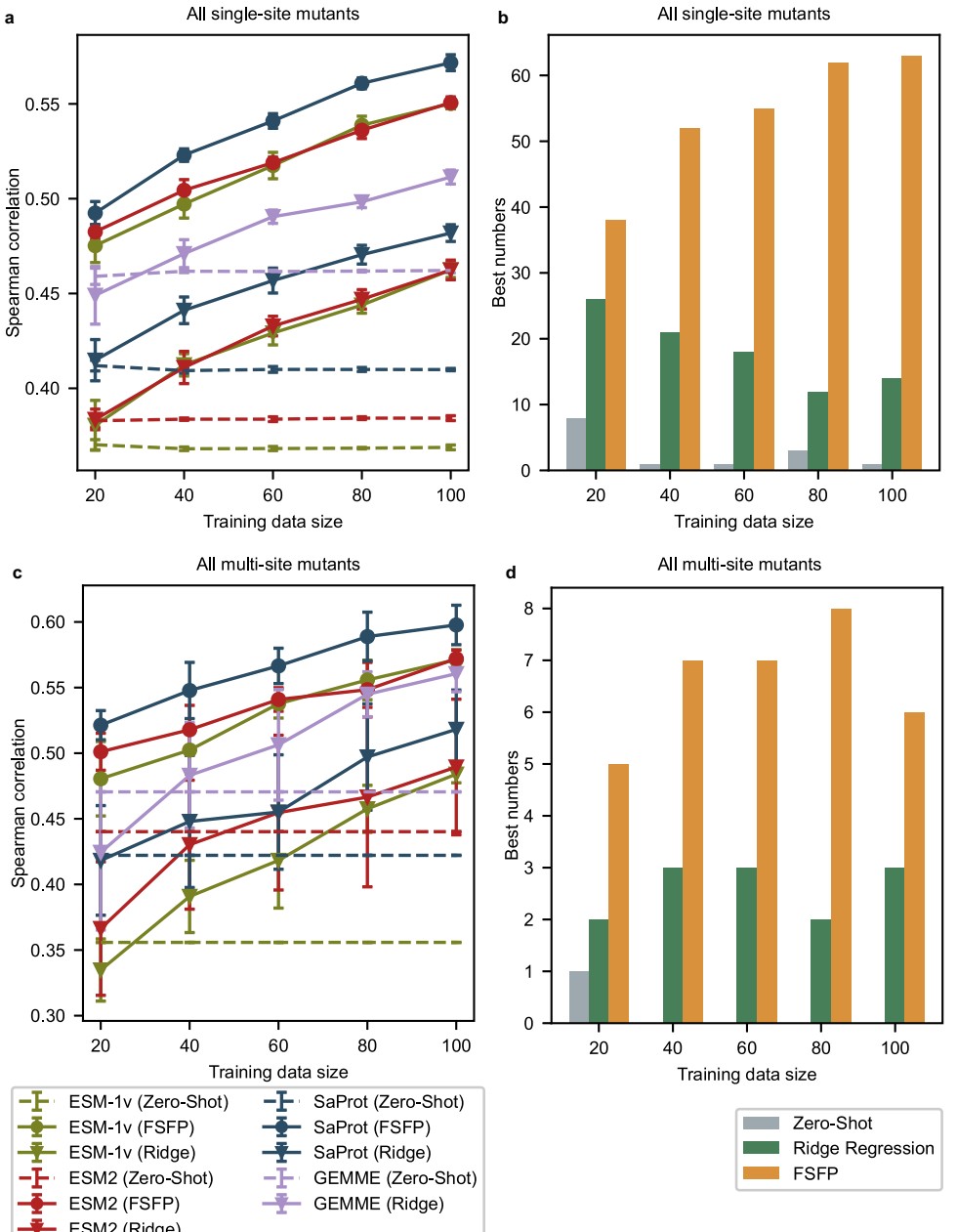

**Fig. 3 | Overall performance on single-site and multi-site mutants. a** Average model performance tested on single-site mutants across all 87 datasets, evaluated by Spearman correlation. Error bars represent the standard deviation caused by five random splits. SaProt (FSFP) is significantly better than all baselines with the largest *P* value among all training sizes being 0.0079 (two-sided Mann–Whitney *U* test). Analogous results measured by NDCG are shown in Supplementary Fig. 3a. **b** Summary of how often the best test Spearman correlation for single-site mutants on a certain dataset is achieved by a PLM, where the colors represent different strategies applied to the best PLMs. **c** Average model performance tested on multi-site mutants across 11 datasets, evaluated by Spearman correlation. Error bars represent the standard deviation caused by five random splits. SaProt (FSFP) is significantly better than all baselines with the largest *P* value among all training sizes being 0.016 (two-sided Mann–Whitney *U* test). Analogous results measured by NDCG are shown in Supplementary Fig. 3b. **d** Similar to (**b**) but counted for the best performance on multi-site mutants. Source data are provided as a Source Data file.

features of the ridge regression approach are much less informative, resulting in limited extrapolation ability.

To further demonstrate the applicability and generalizability of FSFP, we show the comparison results of different approaches on four proteins: the envelope protein Env from HIV[43], the human α-synuclein[44], protein G (GB1)[45], and the human TAR DNA-binding protein 43 (TDP-43)[46]. In these cases, one or more unsupervised models show poor performance and are not reliable in practice, highlighting the need for effective training using limited labeled data (Supplementary Figs. 5 and 6). Notably, for TDP-43, the zero-shot results of all

models have no or even negative correlation with the actual fitness labels (Supplementary Fig. 5d), and those of ESM-2 have low accuracy for HIV Env (Supplementary Fig. 5a) and α-synuclein (Supplementary Fig. 5b). As seen, except for GB1 (Supplementary Fig. 5c), most models enhanced by ridge regression do not show significant improvements against their zero-shot performance, even with larger training datasets. On the contrary, utilizing FSFP, the PLMs achieve considerable gain by training on small datasets, therefore becoming more useful for directed evolution. It can also be observed that the performance of GEMME, the method used by FSFP for yielding pseudo labels, is not dominating

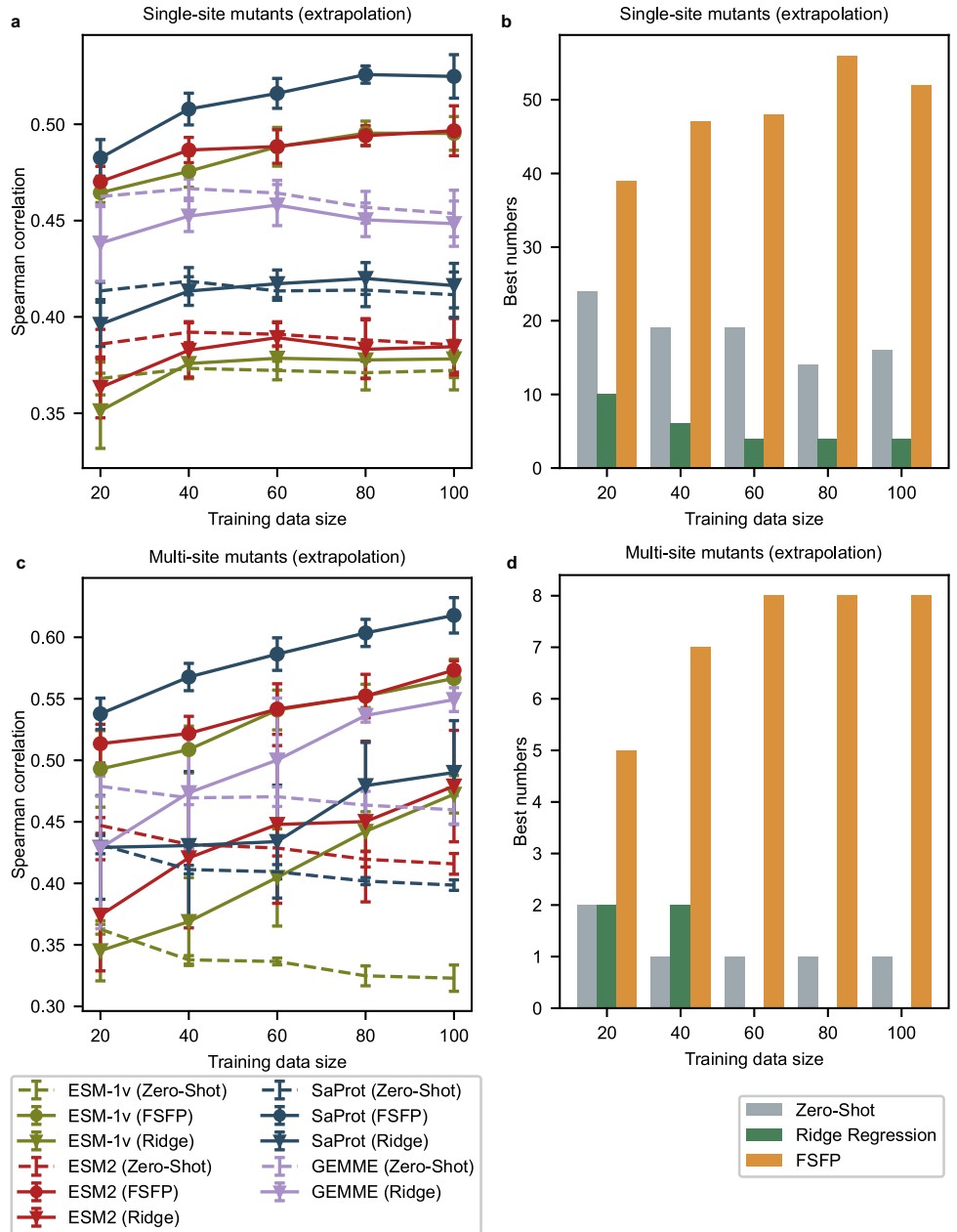

**Fig. 4 | Extrapolative performance on single-site and multi-site mutants.**
**a** Extrapolating to single-site mutants whose mutated positions do not occur in the training set, evaluated by Spearman correlation. Error bars are centered at average performance and indicate the standard deviation caused by five random splits. SaProt (FSFP) is significantly better than all baselines with the largest *P* value among all training sizes being 0.016 (two-sided Mann−Whitney *U* test). Analogous results measured by NDCG are shown in Supplementary Fig. 4a. **b** Summary of how often the best extrapolative Spearman correlation for single-site mutants on a certain dataset is achieved by a PLM, where the colors represent different strategies applied to the best PLMs. **c** Extrapolating to multi-site mutants whose individual mutations have no overlap with the mutations in the training data, evaluated by Spearman correlation. Error bars are centered at average performance and indicate the standard deviation caused by five random splits. SaProt (FSFP) is significantly better than all baselines with the largest *P* value among all training sizes being 0.0079 (two-sided Mann−Whitney *U* test). Analogous results measured by NDCG are shown in Supplementary Fig. 4b. **d** Similar to (**b**) but counted for the best extrapolative performance on multi-site mutants. Source data are provided as a Source Data file.

among other base models in the later three examples. Nevertheless, this does not hinder the FSFP-trained models from being the top performers, which suggests that FSFP learns to generalize from the auxiliary tasks instead of simply overfitting them.

**Engineering of Phi29 using FSFP**
We proceed to demonstrate the practical efficacy of FSFP by engineering the Phi29 DNA polymerase[47] through wet-lab experiments ("Methods"). Phi29 DNA polymerase has a pivotal role in biotechnological applications and has been rigorously validated as an efficient isothermal DNA amplification enzyme[47,48]. Improving the thermostability in Phi29 has currently attracted great research interest[49−51]. Herein, we focus on enhancing its thermostability by starting from acquiring enough positive single-site mutants, so that potentially better multi-site mutants can be originated from them afterward. We apply FSFP to train ESM-1v based on a limited set of wet-lab experimental data and then use it to find new single-site mutants for experimental validation (Fig. 5a and "Methods").

Initially, in the absence of prior wet-lab data, ESM-1v is employed to identify the top 20 single-site mutants of Phi29 based on its zero-

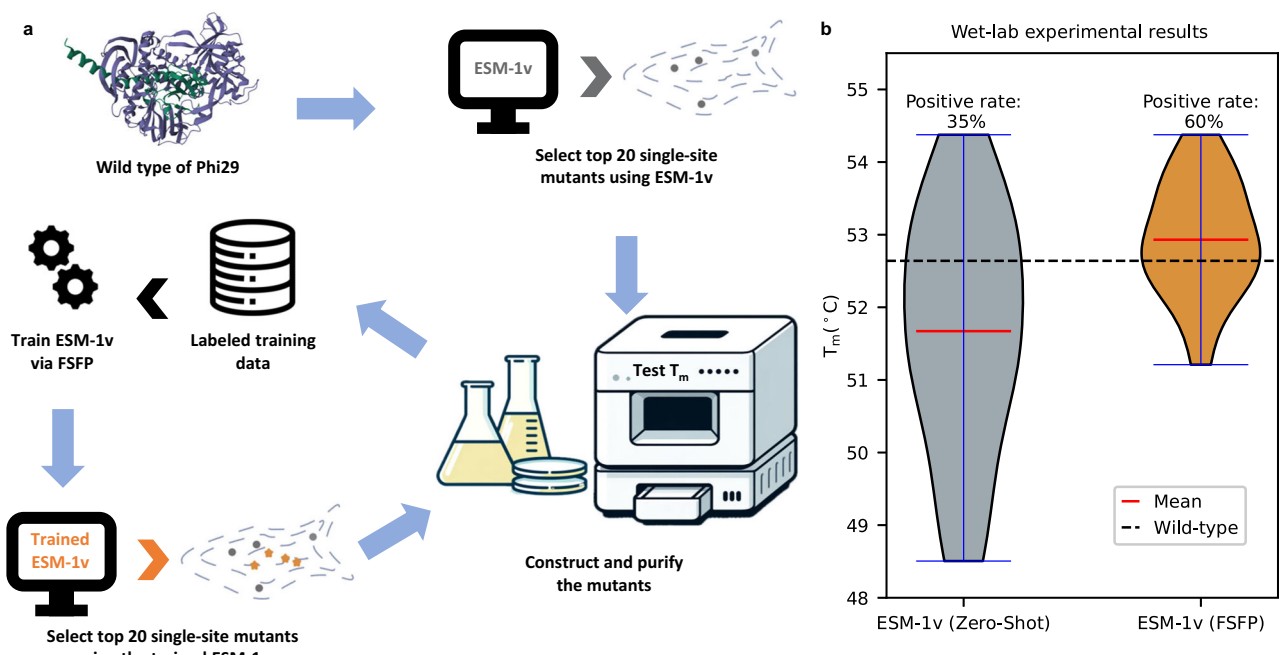

**Fig. 5 | Engineering Phi29 using FSFP. a** The workflow of using FSFP to engineer the Phi29 DNA polymerase. **b** Wet-lab experimental $T_m$ values of the top 20 single-site mutants predicted by ESM-1v before and after training by FSFP. Source data are provided as a Source Data file.

shot predictions for the first round of wet-lab experiments. These mutants are constructed, purified, and subsequently assayed to ascertain their thermal stability. The resultant $T_m$ values are measured and compared against the wild-type baseline. We then train ESM-1v via FSFP on all 20 mutants with these $T_m$ values as labels. The enhanced model is then used to predict a new set of top 20 single-site mutants for further wet-lab experiments.

When comparing the top 20 predictions from ESM-1v before and after FSFP training, it can be found that the average $T_m$ value is improved by more than 1 °C and the positive rate is improved by 25% (Fig. 5b and Supplementary Table 1). Specifically, the best mutant (i.e., the one with the highest $T_m$ value) found by ESM-1v (FSFP) is also recommended by ESM-1v (zero-shot). However, among the positive mutants predicted by ESM-1v (FSFP), nine of them do not appear in the training data, suggesting that FSFP can enable PLMs to identify more protein variants that are better than the wild type. These results affirm the potential of FSFP in accelerating the iterative cycle of design and testing in protein engineering, thereby being helpful to the development of proteins with enhanced functional profiles.

## Discussion

In this work, we introduce FSFP, a paradigm for effectively training PLMs to predict protein fitness using only a small number (tens) of labeled mutants. FSFP integrates the techniques of LTR, LoRA, and MTL, where LTR meets the intrinsic needs of directed evolutions to rank the protein fitness, LoRA greatly reduces the overfitting risk of PLMs when encountering small training datasets, and MTL provides PLMs with better initial parameters for fast adaptation to the target protein. We apply FSFP to three representative PLMs, i.e., ESM-1v, ESM-2, and SaProt for the case study, though it is theoretically compatible with any PLM. Through comprehensive in-silico experiments across 87 DMS datasets, we demonstrate the effectiveness and robustness of FSFP in few-shot protein fitness prediction: (1) it boosts the test performance of the PLMs by up to 0.1 on average Spearman correlation using merely 20 training examples; (2) it improves the performance of different PLMs consistently and considerably; (3) it enables PLMs to well extrapolate to the mutations whose positions are absent in the

training data; and (4) it can be both effective and data-efficient even when the PLMs exhibit poor zero-shot performance on the target protein. We also apply FSFP to engineer Phi29 DNA polymerase using wet-lab experiments and the results show that it significantly improves both the average $T_m$ value and positive rate of the top 20 predictions from ESM-1v.

Reasonably, we find that meta-training PLMs on the proteins that contain more mutants and have higher similarity to the target protein leads to better performance of transfer learning (Supplementary Fig. 7a). Compared with finetuning PLM without MTL (LTR + LoRA), meta-learning is helpful when the dataset size of the auxiliary tasks is ≥500 even if the retrieved proteins have low similarities. Since our third auxiliary task is solely built from the MSA of the target protein, the negative impact of the dissimilar proteins can be mitigated. Notably, in the worst case (the leftmost bar in Supplementary Fig. 7a), the performance of FSFP is comparable to LTR + LoRA and still exceeds zero-shot prediction by a large margin. The underlying reason is that we use the target training data to early stop meta-training ("Methods"), and thus prevent the model from overfitting on the low-quality auxiliary tasks. In general, the more informative the auxiliary tasks for the target protein, the more significant the effect of meta-learning. Therefore, it is important to collect auxiliary datasets that are close enough to the target task based on prior knowledge, e.g., by using different types of experimental data from the target protein.

To search for related proteins, FSFP originally computes the cosine similarities between the protein embeddings yielded by PLMs, while other methods can also be adopted such as MMseqs2[52] and Foldseek[53] (Supplementary Fig. 7b–d). Overall, there is no huge difference in utilizing these search methods, indicating that they are all reliable for identifying relevant training datasets (Supplementary Fig. 7c, d). In addition, we can find that the zero-shot performance of a PLM varies on different types of datasets, e.g., ESM-2 performs best on predicting activity while SaProt performs best on predicting expression (Supplementary Fig. 7c). Such performance trend across the datasets remains after FSFP training, which suggests that FSFP may keep the advantage or bias of the trained PLM over data when boosting its accuracy. Due to the fact that different models suit differently when learning the fitness

landscapes of various proteins (in our experiments, the top-performing model varies for different datasets), the choice of the foundation model to be trained should be given careful consideration.

Based on the superior few-shot performance, FSFP could enable more effective directed evolution, especially when high-throughput screens are difficult. The initial data for directed evolution may originate from rational design, random mutations, or zero-shot predictions from PLMs. Regardless of the proportion of positive mutants in this initial dataset, it could serve as a basis for selecting the most suitable PLMs. In subsequent iterative rounds, FSFP could be applied to train the selected PLMs. Leveraging the extrapolative capabilities of the models trained by FSFP, they can be applied to recommend new mutants.

## Methods

### Efficient solution for the fitness ranking problem

**Parameter-efficient fine-tuning of PLMs.** To prevent PLMs from overfitting on small training datasets, we use LoRA[28] to learn a small number of task-specific parameters instead of optimizing the full model. LoRA hypothesizes that the change of weights during model tuning has a low intrinsic rank. In detail, given a pre-trained weight matrix $\mathbf{W}_0 \in \mathbb{R}^{d \times k}$, LoRA constrains its update by representing the latter with a low-rank decomposition:

$$\mathbf{W}_0 + \Delta\mathbf{W} = \mathbf{W}_0 + \mathbf{BA} \tag{1}$$

where $\mathbf{B} \in \mathbb{R}^{d \times r}$, $\mathbf{A} \in \mathbb{R}^{r \times k}$, and the rank $r \ll \min(d,k)$. $\mathbf{A}$ is randomly initialized but $\mathbf{B}$ is initialized by zero, so $\Delta\mathbf{W}$ is also zero at the beginning of training. We freeze the original pre-trained model and apply LoRA to the weight matrices in its self-attention modules and feed-forward layers. The hyperparameter $r$ is set to 16 in all our experiments.

**Learning to rank the fitness.** We use a listwise LTR approach, namely ListMLE[27] to train PLMs on few-shot training data. It defines the loss function based on the likelihood of the permutation of a correct ranking. Let $\mathbf{x} = \{x_1, \ldots, x_n\}$ be the objects to be ranked, $\mathbf{y} = \{y_1, \ldots, y_n\}$ be their corresponding labels, and $\pi$ be a permutation of $\mathbf{x}$ satisfying $\pi \in \{\pi | y_{\pi(i)} \ge y_{\pi(j)}; i < j\}$ where $\pi(i)$ denotes the index of the object ranked at position $i$ in $\pi$. The ListMLE loss is defined as:

$$\mathcal{L}(f; \mathbf{x}, \pi) = -\log P(\pi | \mathbf{x}; f) \tag{2}$$

where $f$ is a parameterized scoring function, and the probability $P(\pi | \mathbf{x}; f)$ is defined by the Plackett-Luce model[54]:

$$P(\pi | \mathbf{x}; f) = \prod_{i=1}^{n} \frac{\exp\left(f\left(x_{\pi(i)}\right)\right)}{\sum_{k=i}^{n} \exp\left(f\left(x_{\pi(k)}\right)\right)} \tag{3}$$

Given a mutant $\mathbf{s}$, we score its fitness by comparing the probability assigned to each mutated residue by PLMs with the one for the wild type, following Meier et al.[12]:

$$f(\mathbf{s}) = \sum_{t \in T} \left[\log P\left(\mathbf{s}_t = \mathbf{s}_t^{mt} | \mathbf{s}^{wt}\right) - \log P\left(\mathbf{s}_t = \mathbf{s}_t^{wt} | \mathbf{s}^{wt}\right)\right] \tag{4}$$

Here $T$ represents all mutants in $\mathbf{s}$ and $\mathbf{s}_t$ is the residue at position $t$ of the mutant and wild-type sequence. In this way, the initial performance of the supervised model is equal to its zero-shot performance, which makes it easier to fit the training data.

In each training iteration, we randomly select $m$ subsets of size $n$ with replacement from the training data ($n$ is smaller than the training data size, and $m$ can be viewed as batch size) and then compute ListMLE loss on each subset. The average loss on the $m$ subsets is used to update the trainable parameters in PLMs via gradient descent. The

values of $m$ and $n$ depend on the actual training size and the validating performance on the training data.

### Meta-learning on auxiliary tasks

Meta-learning aims to train a model that can quickly adapt to a new task using only a few data points and training iterations, typically by accumulating experience from learning multiple tasks[25,31,32]. Before transferring PLMs to the target few-shot learning task, we perform meta-learning to obtain a better initialization of the LoRA parameters that can learn the target training data faster and further reduce the possibility of overfitting. To build the training tasks required for meta-learning, we search for existing labeled mutant datasets that are potentially helpful to predict the variant effects on the target protein, as well as generate pseudo labels through MSA for the candidate mutants.

**Searching similar experimental datasets.** Since ProteinGym has already included a wide variety of mutant data collected from 87 DMS assays, we use it as a database to search from. First, the wild-type sequences (or structures in the case of SaProt) of the target protein and the ones in the database are sent to a PLM (the one to be trained) to obtain their corresponding embedding vectors. Specifically, the output vectors representing each residue from the last hidden layer are averaged to form the protein embedding vector. Then, the relevance between the target protein and a candidate protein is measured by the cosine similarity between their embeddings $\mathbf{p}$ and $\mathbf{q}$:

$$\text{sim}(\mathbf{p}, \mathbf{q}) = \frac{\mathbf{p} \cdot \mathbf{q}}{\|\mathbf{p}\|_2 \cdot \|\mathbf{q}\|_2} \tag{5}$$

Finally, we choose two proteins that have the highest similarities to the target protein, and take their corresponding DMS datasets as the labeled data for the first two tasks.

**Estimating mutational effects based on MSA.** MSA has proved to be useful in predicting the mutational effects[34-38], and may also boost the performance of PLMs on various downstream tasks[11,18,30]. In this work, we integrate MSA knowledge into PLMs by meta-train PLMs on the pseudo labels generated by an alignment-based method, rather than modifying the model architecture. We choose GEMME algorithm[34] to utilize MSA information, which predicts mutational effects by explicitly modeling interdependencies between all positions in a sequence. It uses MSA to construct Joint Evolutionary Trees[55], and compute conservation degrees of different positions in the protein sequence based on evolutionary traces. The conservation degrees are then used to estimate the evolutionary fit required to accommodate mutations. The relative frequency of the mutation occurrence serves as another quantity to infer the mutational effect. Utilizing GEMME, we score the candidate mutated sequences of the target protein and build the dataset of the third task.

**Meta-training of PLMs.** We apply MAML[32], a state-of-the-art meta-learning algorithm, to enable PLMs to better utilize the few-shot training data. MAML is also successfully applied to recognize drug–target interactions[56] and antigen binding[57]. In essence, MAML learns to find the optimal initial model parameters such that small changes in them will produce large improvements on any target task, when altered by the target loss. Formally, we denote the fixed parameters of the original PLM and the added LoRA parameters as $\boldsymbol{\omega}$ and $\boldsymbol{\theta}$. Before training, for each task $\mathcal{T}_i$ we built, we randomly selected half of its data as training data $\mathcal{D}_i^{tr}$ and the other half as testing data $\mathcal{D}_i^{te}$. Meta-training can be viewed as a bilevel optimization problem. In the inner-level optimization, a task $\mathcal{T}_i$ is randomly picked, and the loss on $\mathcal{D}_i^{tr}$ is used to temporarily update the current trainable parameters $\boldsymbol{\theta}$ into task-specific parameters $\boldsymbol{\theta}_i'$ by gradient descent. Assuming there is only

one gradient update step, it can be expressed as:

$$\boldsymbol{\theta}'_i = \boldsymbol{\theta} - \alpha \nabla_{\boldsymbol{\theta}} \mathcal{L}_{\mathcal{D}_i^{tr}}(f_{\boldsymbol{\omega},\boldsymbol{\theta}}) \qquad (6)$$

Here the loss function $\mathcal{L}$ and scoring function $f$ are the same as the ones in Eq. (2), and $\alpha$ is the step size. For the outer-level optimization, the parameters $\boldsymbol{\theta}$ are trained by optimizing for the test performance of $f_{\boldsymbol{\omega},\boldsymbol{\theta}'_i}$ across the tasks in the current meta-batch:

$$\boldsymbol{\theta} \leftarrow \boldsymbol{\theta} - \beta \nabla_{\boldsymbol{\theta}} \sum_i \mathcal{L}_{\mathcal{D}_i^{te}}\left(f_{\boldsymbol{\omega},\boldsymbol{\theta}'_i}\right) \qquad (7)$$

where $\beta$ is the meta step size. The above two levels of updates are performed repeatedly during the meta-training process until the stop criterion is satisfied. Through meta-learning, we aim to train a meta-learner $f_{\boldsymbol{\omega},\boldsymbol{\theta}}$ that learns to learn similar tasks. After that, we fine-tune $f_{\boldsymbol{\omega},\boldsymbol{\theta}}$ on the target training data and still keep $\boldsymbol{\omega}$ frozen. In our experiments, the gradient step size $\alpha$ and number $g$ for each inner loop are set to 0.005 and 5, respectively, and each meta-batch contains one randomly sampled task. Taking ESM-2 (650 M) as the backbone, we try different combinations of $\alpha$ and $g$ on our benchmark, and find that FSFP is overall not sensitive to $g$ but prefers a smaller $\alpha$ (Supplementary Table 2). Since the outer update of Eq. (7) involves a gradient through a gradient, we use a first-order approximation introduced by Finn et al.[32] to reduce the computational cost. Stochastic gradient descent is applied to solve the inner optimization and we use Adam[58] with default hyperparameters for the outer optimization.

## Foundation models

**ESM1v.** ESM-1v[12] is a transformer language model for variant effects prediction, which employs the ESM-1b architecture and masked language modeling approach of Rives et al.[40]. It is trained on unlabeled protein sequences from Uniref90[59], with 650 M parameters. It includes five models trained with different seeds for ensemble, but we only use the first checkpoint in our experiments.

**ESM-2.** ESM-2[13] introduces improvements to ESM-1b in architecture and increases the data for pre-training. In detail, it equips rotary position embedding[60], and uses UniRef50 for training. We take the 650 M version of ESM-2 for our experiments.

**SaProt.** Su et al. propose SaProt[15], a PLM trained with both protein sequence and structure data. They introduce a structure-aware vocabulary that combines both residue types and 3Di structure tokens encoded by Foldseek. The pre-training dataset of SaProt consists of around 40 M structures predicted by AlphaFold2[61]. SaProt employs the same model architecture as ESM-2, with the structure-aware tokens as input. In this work, we choose the SaProt version that is continuously pre-trained on PDB structures. For SaProt, the scoring function Eq. (4) is modified as follows so that it can fit the new vocabulary:

$$f(\mathbf{s}) = \frac{1}{|\mathcal{U}|} \sum_{t \in T} \sum_{u \in \mathcal{U}} \left[ \log P\left(\mathbf{s}_t = \mathbf{s}_t^{mt} u | \mathbf{s}^{wt}\right) - \log P\left(\mathbf{s}_t = \mathbf{s}_t^{wt} u | \mathbf{s}^{wt}\right) \right] \qquad (8)$$

where $u$ is a 3Di token, $\mathcal{U}$ is the structure alphabet used by Foldseek, and $\mathbf{s}_t u$ is a structure-aware token in the new vocabulary.

## Early stopping strategy

Early stopping is widely used in deep learning to prevent the model from overfitting. When sufficient labeled data are available, it is generally based on a separate validation set. However, a held-out validation set may result in insufficient training data in a low-resource scenario. On the other hand, if the validation data size is assigned too small, the validation scores such as Spearman correlation may not be representative enough for early stopping. Based on these

considerations, we propose to estimate the number of training iterations for transfer learning by Monte Carlo cross-validation[62]. Specifically, we create five random splits of the training set into training and validating data. The proportion of training and validating data is 0.5:0.5 when the training data size is <50 otherwise 0.75:0.25. For each split, the model is trained on the sub-sampled training data for up to 500 steps, and we record the Spearman correlation calculated on the validating data every five steps. After five rounds of training and validating, we choose the training step number with the highest average validation score across different splits, and finally train the model for that number of steps on the whole training data.

The training data from the target protein is also used to early stop the meta-training procedure. Similarly, five random splits of it are generated first. For every five steps of the outer optimization during meta-training, we train the current meta-learned model $f_{\boldsymbol{\omega},\boldsymbol{\theta}}$ on the sub-sampled target training sets for five gradient steps (same to the inner optimization) and compute the validation Spearman scores. We stop meta-training if the average validation score of different splits does not improve within 20 consecutive records and pick the best meta-learner according to this score.

## Benchmark datasets

ProteinGym is an extensive set of DMS assays for comparisons of different mutational effect predictors[30]. We evaluate the model performance on its substitution benchmark, which consists of about 1.5 M missense variants from 87 DMS assays. Since the maximum input length of ESM-1v is 1024 tokens, we truncate the proteins that have more than 1024 amino acids and ensure that most of the mutations in their corresponding datasets occur in the resulting interval. To build the inputs for SaProt, we obtain the structures of the proteins via AlphaFold2 or download them from AlphaFoldDB if available.

For each dataset in ProteinGym, we first randomly sample 20 single-site mutants as an initial training set. On the basis of these training examples, we add another 20 randomly selected single-site mutants to build the second training set. Analogously, we expand the training set size to 60, 80, and 100 for separate experiments. For each training set, all the remaining data (or part of them if we specify) are used as test set and we do not access them for hyperparameter selection. The above process for data splitting is repeated five times with different random seeds, and we report the average model performance for each training data size. Since the test data changes across the experiments, the zero-shot performance is not constant.

## Protocol of wet-lab experiment for Phi29

Phi29 DNA polymerase is extensively employed in various DNA amplification techniques, such as rolling-circle amplification, multiple displacement amplification (MDA), and non-specific whole genome amplification (WGA). It has been rigorously validated as an efficient isothermal DNA amplification enzyme[47,48]. Each amplification method necessitates distinct enzymatic characteristics. For example, WGA and MDA reactions performed at elevated temperatures could benefit from expedited reaction kinetics. Additionally, increased reaction temperatures may mitigate the influence of C/G content on amplification bias, offering further advantages[49,50]. Consequently, the enhancement of thermostability in Phi29 has been a focal point of academic research. Despite extensive documentation of engineered protein, the thermal stability of these variants has yet to meet the practical application requirements, with no substantial breakthroughs reported to date[49,51,63].

**Mutant selection procedure.** Initially, we use an ensemble of five ESM-1v models to score saturated single-site mutants of Phi29, where the zero-shot predictions of five models are averaged. Then, 20 mutants with the highest predicted scores are chosen to measure experimental $T_m$ values. Subsequently, a training dataset comprising these mutants

with the obtained $T_m$ values as labels is constructed. FSFP is then applied to train the first ESM-1v model in the ensemble to refine its predictions. The top 20 predictions from the trained model for single-site mutants are selected for the next iteration of wet-lab experiments.

**Plasmid construction.** A codon-optimized version of Phi29 DNA polymerase and variants genes is synthesized by Sangon Biotech (Shanghai, China). It is cloned into the pET28(a) plasmid (the construction of Phi29 protein is shown in Supplementary Fig. 8) to construct pET28a-phi 29-MX with an N-terminal His-tag.

**Protein expression.** The expression plasmid is transformed into *Escherichia coli* BL21(DE3) cells. A 30 ml seed culture is grown at 37 °C in LB medium with 50 μg/ml kanamycin and is subsequently transferred to 500 mL of LB in a shaker flask containing 50 μg/ml kanamycin. The cultures are incubated at 37 °C until the OD600 reaches 1.0, and protein expression is then induced by the addition of isopropyl-D-thiogalactopyranoside to a final concentration of 0.5 mM, followed by incubation for 16 h at 25 °C.

**Protein purification.** Cells are harvested by centrifugation for 30 min at $3345 \times g$ and the cell pellets are collected for later purification. The cell pellets are resuspended in lysis buffer (25 mM Tris-HCl, 500 mM NaCl, pH 7.4) and then disrupted using ultrasonification (Scientz, China). The lysates are centrifuged for 30 min at $15,615 \times g$ at 4 °C, after which the supernatants are subjected to Ni-NTA affinity purification with elution buffer (25 mM Tris-HCl, 500 mM NaCl, 250 mM imidazole, pH 7.4). Further, the protein is desalted with lysis buffer (25 mM Tris-HCl, 500 mM NaCl, pH 7.4) using ultrafiltration. The fractions containing the protein are flash frozen at −20 °C in buffer (25 mM Tris-HCl pH 7.4, 200 mM NaCl, 20% glycerol).

**Melting temperature assessment.** The $T_m$ values are determined by differential scanning fluorimetry (DSF) method using the Protein Thermal Shift Dye Kit (Thermo Fisher). One microlitter of the SSYPRO Orange dye (SUPELCO, USA) is added to 49 μL lysis buffer (25 mM Tris-HCl, 500 mM NaCl, pH 7.4). Then, 1 μL diluted dye is mixed with 19 μL of 0.1 mg/mL protein. DSF experiments are then carried out using the LightCycler 480 Instrument II (Roche, USA). The reaction mixture first reaches 25 °C, then raises to 99 °C at the speed of 0.05 °C/s and maintains it for 2 min. Protein Thermal Shift software is used for data processing.

### Reporting summary
Further information on research design is available in the Nature Portfolio Reporting Summary linked to this article.

## Data availability
The datasets used for benchmarking are from ProteinGym[30] https://github.com/OATML-Markslab/Tranception. The detailed performances of different approaches on each dataset measured by Spearman correlation and NDCG are available in Supplementary Data 1 and 2, respectively. Source data are provided with this paper.

## Code availability
The source code of FSFP is available at https://github.com/ai4protein/FSFP.

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

## Acknowledgements

This work was supported by the grants from the National Science Foundation of China (Grant Number 12104295), the Computational Biology Key Program of Shanghai Science and Technology Commission (23JS1400600), Shanghai Jiao Tong University Scientific and Technological Innovation Funds (21X010200843), and Science and Technology Innovation Key R&D Program of Chongqing (CSTB2022TIAD-STX0017), the Student Innovation Center at Shanghai Jiao Tong University, and Shanghai Artificial Intelligence Laboratory.

## Author contributions

P.T. and L.H. conceptualized and supervised this research project. Z.Z. and P.T. developed the methodology and designed the benchmark. Z.Z., L.Z., and Y.Y. implemented the method. Z.Z., L.Z., Y.Y., and M.L. performed the computational experiments and analyzed the results. B.W. conducted the wet-lab experiments of Phi29. Z.Z., Y.Y., B.W., and P.T. wrote the manuscript. Z.Z., L.Z., Y.Y., and B.W. contributed equally to this work. All authors reviewed and accepted the manuscript.

## Competing interests

A patent application 2024106909647 relating to the mutants of the Phi29 DNA polymerase developed in this study has been filed in the name of Matwings Technology Co., Ltd., pending. B.W. and P.T. are the inventors of this patent. The other authors declare no competing interests.
