## [Peer Review File · Nature Communications]

Enhancing the efficiency of protein language models with minimal wet-lab data through few-shot learningREVIEWER COMMENTS

Reviewer #1 (Remarks to the Author):

Accurately predicting protein fitness is essential for protein engineering. Currently the experimental technique can produce data in a large scale but is expensive and slow, and computational methods are urgently required. This study employed meta-learning to train the models and achieved superior results. The idea is quite direct and reasonable. The writing is clear and easy to follow. However, the currently presented form needs large improvements and deep investigation of the underlying mechanism is required.

Major:

1. The ridge-regression was currently used as baseline but this is the fitting model, not in the same level as MAML. Usually, a multi-task technique is used for showing the efficiency.
2. The manuscript is limited in SPC comparison. Since SPC might be biased, a direct comparison of predicted vs actual values is required.
3. MAML is well known to be sensitive to the hyper-parameters, and the parameter sensitivity should be analyzed.
4. For the specific cases, the authors need to dive into the reasons why the current strategy works, and what is the strength and weakness of current models: e.g. any bias on amino acid types, secondary structure, etc. Do the prediction depend on homology and what's the performance if excluding homologous protein during the meta-training.
5. As the authors' group has wet experimental conditions, they are suggested to prove their method on one case to improve the affinity. If they could validate their predictions by engineering the proteins, their work will be greatly improved.
6. Literature review: the authors should review related works using meta-learning, e.g. protein engineering, drug optimization.
7. Most figures needs re-designing to avoid the large blank regions.

Reviewer #2 (Remarks to the Author):

The authors present a new protocol for predicting the fitness landscape of a target protein. Towards this, they build on top of pre-trained language models which they finetune via LORA, a parameter-efficient finetuning technique. Additionally, they leverage existing experimental information from different yet related proteins (meta-transfer learning) and instead of predicting the change of a protein's fitness upon mutation numerically, the authors propose to instead simply rank mutants which bypasses the problem of enforcing different protein functions/fitnesses on the same scale. The authors evaluate the model using the established protein gym dataset which is a cleaned and pre-processed set of 87 deep mutational scanning assays.

The authors make their method available and perform an ablation study to understand the impact on individual components. The paper is well written but the MAML description could be improved to bring readers up to speed who are not familiar with the concept. I would still like to better understand why training on potentially unrelated proteins with vastly different sequence/structure/function helps so much in predicting the mutation effect of a target protein. Especially, as this appears to be the major driver for performance improvement on extrapolation (Fig. 4a).

Therefore, I wonder mostly about the relationship between the choice of related proteins and performance. When choosing two proteins from protein gym to get additional data, I think it would be interesting to better understand the effect of a) how to choose the proteins, i.e., did you compare embedding distance to sequence similarity (MMSeqs2) or structure similarity (Foldseek)? - Also, irrespective of the method chosen for picking those additional proteins, it would be interesting to evaluate the impact of similarity between those proteins and the actual target protein. E.g., is there any relationship between the similarity of the related proteins to the target protein? (assumption would be: the more similar the related proteins for which you already have

experimental data, the more informative for the target protein). - This ties in with your Discussion statement about "the similarities between the proteins in ProteinGym are low overall". Would suggest quantifying. The same goes for the number of mutations available for the specific choice of related proteins, i.e., does the model perform better if more mutations are provided via related proteins?

Fig. 1: make more clear what objective/loss is used in panel B (similar to panel C where you state that you use ranking loss).

Fig. 1: similarly, make clear whether only LORA adapters are finetuned in panel C or the full model.

Introduction: you state ESM-IF as an example for pLMs. However, this model is an inverse folding model which has a different objective and requires labelled data (3D structures). There are many more publicly available (s.a. ProtGPT2, ProtT5, SeqVec to name a few), open source pLMs. I would suggest to rather reference one of those or simply remove the ESM-IF citation.

You state that accuracy of pLMs remains limited. While this is correct, esp. w.r.t. Mutation effect prediction, this sounds a bit too negative to me. Sure there is plenty of room for improvement but after all, most high-scoring methods for protein-gym are either pLM based approaches or hybrid approaches (see public leaderboard at proteingym.org shows that TranceptEVE, EVE, VESPA are among the best methods and those are all hybrid/pLM-based).

Results: "ProteinGym is used as the database to retrieve due to the diversity". I am not sure whether I would argue about diversity if you are working with a set of 87 proteins.

There is already existing work that uses GEMME to generate auxiliary labels; maybe reference: "Alignment-based protein mutational landscape prediction doing more with less"

The concept of MAML might not be widely known. Expanding the brief introduction at the beginning of Results might help to bring readers up to speed. Maybe also adding an example helps.

Results: "MSE denotes fine-tuning the entire PLM with fitness labels as done by Rives et al.,". I thought that Rives et al. simply took the output probability of the amino acids (or rather the difference thereof) for approximating mutation effect, i.e., there was no finetuning involved from what I remember. Maybe double check and clarify/rewrite if needed.

Results: "Meanwhile, LTR + LoRA outperforms the above methods on all training set sizes in terms of both Spearman correlation and NDCG". Make clear that this is a bit circular: you replace MSE by ranking-loss, benchmark via metrics that solely care about ranking and you do better. This is fine but it should be stated very clearly.

Fig 2 a and 2b show identical trends. Adding panel B only adds minor information. Would consider moving to SOM.

Discussion: " For example, when predicting the properties highly correlated to protein structure such as binding and thermostability, one can select SaProt". Any way to back this up with numbers/examples?

Add in main text which ESM version was used. Consider benchmarking other versions of ESM as well. It would be interesting to see the effect of model size on your benchmark, i.e., does model size really constantly lead to better downstream performance?

How were structures computed for SAProt?

Given that LORA injects adapters in each layer, the architecture of the underlying pLMs can have an impact on how many trainable parameters each model has at the end. State clearly how many trainable parameters each of the compared models has.

Make absolutely clear that protein gym compares methods that never used any DMS data for training while you did. Once you did this, you could also put your method into perspective of fully unsupervised methods compared in the public leaderboard (of course, always with the remark of the difference between unsupervised vs supervised approach).

- I appreciate that you visualized all your results but consider adding a table with numbers and standard errors to SOM to have exact numbers to compare (might be important for others in the future who would like to compare to your method).

I hope my comments above provide some constructive feedback - I really like the idea and generality of your approach - best,
Michael Heinzinger

Reviewer #2 (Remarks on code availability):

Did not run the code but the README of the github suggests that reported performances can be reproduced. Given the large numbers of parameters available, I suggest to provide a script that allows to reproduce the numbers reported in the paper.

Response to the Referees for manuscript NCOMMS- 24-07326

Response to Reviewer1

- 1. The ridge-regression was currently used as baseline but this is the fitting model, not in the same level as MAML. Usually, a multi-task technique is used for showing the efficiency.**

Response: We thank Reviewer1 for the comments and suggestions. In the ablation study, we have compared ridge-regression with the non-MAML version of FSFP, i.e., LTR + LoRA (Fig.R1 below). The results show that LTR + LoRA already outperforms ridge-regression while MTL further improves the performance of our approach.

However, we find that directly applying multi-task learning may be neither a suitable solution to few-shot protein fitness prediction nor a proper baseline in such scenario. Conceptually, multi-task learning and meta-learning aim to solve different problems despite that they both leverage the correlation between the training tasks. Multi-task learning aims to learn shared representations to *improve the performance on all training tasks simultaneously* (Zhang et al., A survey on multi-task learning, IEEE Transactions on Knowledge and Data Engineering, 34(12), 2022), while the goal of meta-learning is *fast adaptation to unseen task* (Huisman et al., A survey of deep meta-learning, Artificial Intelligence Review, 54, 2021). Due to its problem formulation, multi-task learning does not allow adaptation to unseen tasks, i.e., generalize from auxiliary tasks to the target task, at least in a straightforward manner.

To apply multi-task learning to our scenario, a PLM should be simultaneously trained on the target task along with the auxiliary tasks we built, and the performance on them is expected to be jointly improved. Unfortunately, the training data size of these tasks are extremely unbalanced, e.g., tens of training examples for the target task and thousands of those for the auxiliary tasks, and thus the resulted model will be biased towards the auxiliary tasks and not suitable for the target protein. On contrary, MAML enables the PLM to learn optimal initial parameters from the auxiliary tasks so that it can rapidly adapt to the target task. We have tried our best to research the literature but failed to find effective implementation of multi-task learning for few-shot protein fitness prediction. If available, we welcome Reviewer1 to recommend a proper multi-task learning baseline for our scenario.

Fig.R1 Ablation study on ESM-2 (copied from Fig.2 in the manuscript). **a)** Average performance of each strategy across all datasets in ProteinGym with respect to the training data size, evaluated by Spearman correlation. For each dataset, we randomly pick 20, 40, 60, 80, and 100 single-site mutants as training set, with all the rest as test set. Each dot in the figure is the average test performance of 5 random data splits, and the error bars represent the standard deviation caused by different splits. Two-sided Mann-Whitney U -tests are used for comparing the performance of FSFP with all other strategies, getting $P < 0.0079$ for all training sizes. **b)** Distribution of the performance improvement in Spearman correlation over zero-shot prediction across all datasets in ProteinGym, with the training size of 40. The performance gain of each dataset is averaged among the 5 random splits.

2. The manuscript is limited in SPC comparison. Since SPC might be biased, a direct comparison of predicted vs actual values is required.

Response: Thanks for the advice. Generally, the goal of protein engineering is to identify mutants with enhanced fitness, so ranking-related metrics assume greater importance. Following the comment, we have further added two metrics that take the actual label values into account in ablation study: Pearson correlation and MAE (Fig.R2 below and Page 5 in the revised manuscript). FSFP effectively improves the performance of ESM-2 in terms of Pearson correlation but cannot optimize for MAE. Meanwhile,

Ablation study on ESM-2 (87 datasets)

Fig.R2 Ablation study on ESM-2 (copied from Fig.S1 in the SI). **a)** Effect of changing the model size, with the training set size of 40. The performance is averaged across all datasets in ProteinGym, and error bars represent the standard deviation caused by random splits. The 650M model is chosen for other experiments. Average performance of different strategies is evaluated by **b)** NDCG, **c)** Pearson correlation, and **d)** MAE. When calculating MAE, the labels in the test set are standardized by removing the mean and scaling to unit variance.

the regression-based methods exhibit substantial MAE as well (Fig.R2d), rendering their absolute output values also impractical for real-world applications. In the context of few-shot learning, predicting exact label values becomes challenging due to significant differences in their range between training and testing data. Therefore, we recommend using LTR to train PLMs, which aligns with the objective of improving mutant fitness rather than predicting precise numerical values.

3. MAML is well known to be sensitive to the hyper-parameters, and the parameter sensitivity should be analyzed.

Response: Thanks for this insightful comment. The gradient step size α and number g for each inner loop of MAML are important hyperparameters. Taking ESM-2 (650M) as the backbone, we have tried different combinations of α and g on our benchmark and find that FSFP is overall not sensitive to g but prefers a smaller α (Table R1 below and Page 14 in the revised manuscript). As shown in Fig.R2a, we have also analyzed the impact of model size on FSFP, and find that FSFP keeps achieving better performance on larger model (Page 5 in the revised manuscript).

Table R1. Performance of ESM-2 (FSFP) under different MAML settings (copied from Table S2 in the SI).

MAML setting		Spearman correlation
$\alpha = 0.005$	$g = 2$	0.503 ± 0.004
	$g = 3$	0.503 ± 0.001
	$g = 4$	0.500 ± 0.003
	$g = 5$	0.503 ± 0.006
	$g = 6$	0.500 ± 0.006
$g = 5$	$\alpha = 0.1$	0.487 ± 0.008
	$\alpha = 0.05$	0.490 ± 0.007
	$\alpha = 0.01$	0.496 ± 0.004
	$\alpha = 0.005$	0.503 ± 0.006
	$\alpha = 0.001$	0.503 ± 0.004

Average performance across all datasets in the benchmark are reported, along with the standard deviation caused by random splits. The training set size is 40. α and g is the gradient step size and number during the inner loop of MAML.

4. For the specific cases, the authors need to dive into the reasons why the current strategy works, and what is the strength and weakness of current models: e.g. any bias on amino acid types, secondary structure, etc. Do the prediction depend on homology and what's the performance if excluding homologous protein during the meta-training.

Response: Thank Reviewer1 for these suggestions. We have added an experiment where we deliberately limit the number of mutants in the auxiliary tasks and take the labeled data from dissimilar proteins, i.e., with the lowest similarities to the target protein (Fig.R3 below and Page 12 in the revised manuscript). Reasonably, we find that meta-training PLMs on the proteins that contain more mutants and have higher similarity to the target protein leads to better performance of transfer learning (Fig.R3a). Compared with finetuning PLM without MTL (LTR + LoRA), meta-learning is helpful when the dataset size of the auxiliary tasks is larger than 500 even if the retrieved proteins have low similarities. Since our third auxiliary task is solely built from MSA of the target protein, the negative impact of the dissimilar proteins can be mitigated. Notably, in the worst case (the leftmost bar in Fig.R3a), the performance of FSFP is comparable to LTR + LoRA and still exceeds zero-shot prediction by a large margin. The underlying reason is that we use the target training data to early stop meta-training (Methods), and thus prevent the model from overfitting on the low-quality auxiliary tasks. In general, the more informative the auxiliary task for the target protein, the more significant the effect of meta-learning.

The quality of the pseudo labels generated by GEMME can also affect the performance of FSFP. If the number of homologous sequences after alignment is quite limited, the model will fail to learn useful information from the third auxiliary task. In Fig.R1, we have shown that the model performance drops if we exclude MSA information during meta-learning (FSFP v.s. LTR + LoRA + MTL (no MSA)), but

Fig.R3 Comparison of different auxiliary task selection strategies (copied from Fig.S7 in the SI). **a)** Performance of FSFP when limiting the number of mutants in the auxiliary tasks and (or) taking the labeled data from dissimilar proteins (i.e., with the lowest similarities to the target protein). The base model is ESM-2 and the target training set size is 40. Error bars represent the standard deviation caused by random splits. **b)** Similarity of the most relevant protein retrieved for building auxiliary tasks, using MMseqs2 and FoldSeek respectively. **c)** Break-down performance by the function to predict, using different methods to search similar proteins. The target training set size is 40. **d)** Similar to Fig.R3c, but performance is by the taxon of the target protein.

still better than the zero-shot inference.

We have also explored the effect of using MMseqs2 (considering the sequence identity) and FoldSeek (considering the structure similarity) to search for related proteins, and reported the performance of different approaches by the taxon and function of the target protein (Fig.R3b,c,d and Page 12 in the revised manuscript). Overall, there is no huge difference in utilizing these search methods, indicating that they are all reliable for identifying relevant training datasets (Fig.R3c,d). In addition, we can find that the zero-shot performance of a PLM varies on different types of datasets, e.g., ESM-2 performs best on predicting activity while SaProt performs best on predicting expression (Fig.R3c). Such performance trend across the datasets remains after FSFP training, which suggests that FSFP may keep the advantage or bias of the trained PLM over data when boosting its accuracy.

Overall, the *strength* of FSFP is that in most cases, e.g., utilizing different protein search methods, it can significantly boost the performance of PLMs using few labeled mutants of the target proteins. Its *weakness* is that the improvements brought by FSFP can be affected by the quality of the built auxiliary tasks as well as the inductive bias of the trained PLMs. The above discussion has been summarized in the revised manuscript.

5. As the authors' group has wet experimental conditions, they are suggested to prove their method on one case to improve the affinity. If they could validate their predictions by engineering the proteins, their work will be greatly improved.

Response: We have demonstrated the practical efficacy of FSFP by engineering the Phi29 DNA polymerase using wet-lab experiments (Page 10, 11 in the revised manuscript). Phi29 DNA polymerase has a pivotal role in biotechnological applications, and has been rigorously validated as an efficient isothermal DNA amplification enzyme. Improving the thermostability in Phi29 is important for its application during efficient isothermal DNA amplification and thus has currently attracted great research interest (Ordóñez et al., International Journal of Molecular Sciences 24, 9331 (2023); Povilaitis et al., Protein Engineering, Design Selection 29, 617-628 (2016)). Herein, we focused on enhancing its thermostability by starting from acquiring enough positive single-site mutants, so that potentially

Fig.R4 Engineering Phi29 using FSFP (copied from Fig.5 in the manuscript). a) The workflow of using FSFP to engineer the Phi29 DNA polymerase. b) Wet-lab experimental T_m values of the top 20 single-site mutants predicted by ESM-1v before and after training by FSFP.

better multi-site mutants can be originated from them afterwards. We applied FSFP to train ESM-1v based on a limited set of wet-lab experimental data and then used it to find new single-site mutants for wet-lab experiments (Fig.R4a and Methods).

Initially, in the absence of prior wet-lab data, ESM-1v was employed to identify the top 20 single-site mutants of Phi29 based on its zero-shot predictions for the first round of wet-lab experiments. These mutants were constructed, purified, and subsequently assayed to ascertain their thermal stability. The resultant melting temperatures (T_m) were measured and compared against the wild-type baseline. We then trained ESM-1v via FSFP on all 20 mutants with these T_m values as labels. The enhanced model was then used to predict a new set of top 20 single-site mutants for further wet-lab experimental evaluation.

When comparing the top 20 predictions from ESM-1v before and after FSFP training, it can be found that the average T_m value is improved by more than 1°C and the positive rate is improved by 25% (Fig.R4b and Table R2). Among the positive mutants predicted by ESM-1v (FSFP), 9 of them do not appear in the training data, suggesting that FSFP can enable PLMs to identify more protein variants with higher fitness. These results affirm the potential of FSFP in accelerating the iterative cycle of design and testing in protein engineering, thereby being helpful to the development of proteins with enhanced functional profiles. We have made the checkpoint for the trained ESM-1v available on the public GitHub repository.

Table R2. Wet-lab experimental T_m for the single-site mutants of Phi29 (copied from Table S1 in the SI).

From ESM-1v (Zero-Shot)	T_m(°C)	From ESM-1v (FSFP)	T_m(°C)
T441L	54.38	T441L	54.38
S10I	53.75	Q55D	53.94
G245V	53.46	S551L	53.90
Q257I	53.37	V19P	53.86
V130L	53.09	L567E	53.61
P129S	52.82	G245V	53.46
V54N	52.67	V566E	53.38
Wild-type	52.64	V130L	53.09
C290K	52.58	S551M	53.04
Q257V	51.97	H3K	52.92
Q257A	51.97	F526L	52.84
W367R	51.74	T140P	52.76
Q257L	51.27	Wild-type	52.64
Y449G	51.11	C290K	52.58
V54E	51.04	P558W	52.56
M30Y	51.03	V566K	52.50
Y369E	50.51	V568K	52.40
W327D	49.90	Y224D	52.23
C530K	49.30	P404E	52.20
H35G	48.97	M506T	51.76
W327K	48.51	T542Y	51.21

The mutants are the top 20 predictions from ESM-1v before and after trained by FSFP respectively.

6. Literature review: the authors should review related works using meta-learning, e.g. protein engineering, drug optimization.

Response: Thanks for reminding. Two references have been added to Page 14 in the revised manuscript:

Wang et al., ZeroBind: a protein-specific zero-shot predictor with subgraph matching for drug-target interactions, *Nature Communications*, 14, 2023

Gao et al., Pan-Peptide Meta Learning for T-cell receptor–antigen binding recognition, *Nature Machine Intelligence*, 5, 2023

7. Most figures need re-designing to avoid the large blank regions.

Response: We have removed the blank regions in the figures. Thanks for pointing this out

Response to Reviewer2

1. When choosing two proteins from protein gym to get additional data, I think it would be interesting to better understand the effect of a) how to choose the proteins, i.e., did you compare embedding distance to sequence similarity (MMSeqs2) or structure similarity (Foldseek)?

Response: We thank Reviewer2 for the comments and suggestions. We have explored the effect of using MMSeqs2 and FoldSeek to search for related proteins, and report the performance of different approaches by the taxon and function of the target protein (Fig.R3b,c,d in the first reviewer’s comment). Overall, there is no huge difference in utilizing these searching methods, indicating that they are all reliable for identifying relevant training datasets (Fig.R3c,d). In addition, we can find that the zero-shot performance of a PLM varies on different types of datasets, e.g., ESM-2 performs best on predicting activity while SaProt performs best on predicting expression (Fig.R3c). Such performance trend across the datasets remains after FSFP training, which suggests that FSFP may keep the advantage or bias of the trained PLM over data when boosting its accuracy.

2. Also, irrespective of the method chosen for picking those additional proteins, it would be interesting to evaluate the impact of similarity between those proteins and the actual target protein. E.g., is there any relationship between the similarity of the related proteins to the target protein? This ties in with your Discussion statement about “the similarities between the proteins in ProteinGym are low overall”. Would suggest quantifying. The same goes for the number of mutations available for the specific choice of related proteins, i.e., does the model perform better if more mutations are provided via related proteins?

Response: We thank Reviewer for the questions. We have added an experiment where we deliberately limit the number of mutants in the auxiliary tasks and take the labeled data from dissimilar proteins, i.e., with the lowest similarities to the target protein (Page 12 in the revised manuscript and Fig.R3a in the first reviewer’s comment). Reasonably, we find that meta-training PLMs on the proteins that contain more mutants and have higher similarity to the target protein leads to better performance of transfer learning (Fig.R3a). Compared with finetuning PLM without MTL (LTR + LoRA), meta-learning is helpful when the dataset size of the auxiliary tasks is larger than 500 even if the retrieved proteins have low similarities. Since our third auxiliary task is solely built from MSA of the target protein, the negative impact of the dissimilar proteins can be mitigated. Notably, in the worst case (the leftmost bar in Fig.R3a), the performance of FSFP is comparable to LTR + LoRA and still exceeds zero-shot prediction by a large margin. The underlying reason is that we use the target training data to early stop meta-training (Methods), and thus prevent the model from overfitting on the low-quality auxiliary tasks. In

general, the more informative the auxiliary task for the target protein, the more significant the effect of meta-learning.

We have removed the statement about “the similarities between the proteins in ProteinGym are low overall” and instead provided the similarity distribution of the most relevant protein retrieved for building auxiliary tasks (Fig.R3b).

3. Fig. 1: make more clear what objective/loss is used in panel B (similar to panel C where you state that you use ranking loss).

Response: We thank Reviewer for pointing this out. During MAML training, ranking loss is used. We have updated Fig.1b in the revised manuscript as below.

4. Fig. 1: similarly, make clear whether only LORA adapters are finetuned in panel C or the full model.

Response: When transferring PLMs to the target task, only LoRA parameters are updated. We have updated Fig.1c as below.

5. Introduction: you state ESM-IF as an example for pLMs. However, this model is an inverse folding model which has a different objective and requires labelled data (3D structures). There are many more publicly available (s.a. ProtGPT2, ProtT5, SeqVec to name a few), open source pLMs. I would suggest to rather reference one of those or simply remove the ESM-IF citation.

Response: We thank Reviewer2 for pointing us to the more suitable references. We have replaced the ESM-IF citation with ProtT5 (Page 2 in the revised manuscript):

Elnaggar et al., ProtTrans: Toward understanding the language of life through self-supervised learning, IEEE Transactions on Pattern Analysis and Machine Intelligence, 44(10), 2022

-
- 6. Results: “ProteinGym is used as the database to retrieve due to the diversity”. I am not sure whether I would argue about diversity if you are working with a set of 87 proteins.**

Response: We have rewritten this statement to “ProteinGym is used as the database to retrieve because it was the largest public collection of DMS datasets at the time of writing.”

- 7. There is already existing work that uses GEMME to generate auxiliary labels; maybe reference: “Alignment-based protein mutational landscape prediction doing more with less”**

Response: We thank Reviewer2 for pointing us to this paper. We have added this reference to Page 4, 8 and 14 in the revised manuscript.

- 8. The concept of MAML might not be widely known. Expanding the brief introduction at the beginning of Results might help to bring readers up to speed. Maybe also adding an example helps.**

Response: We have slightly improved the description of MAML on Page 4 in the revised manuscript: We apply model-agnostic meta-learning (MAML), a popular gradient-based meta-learning method, to meta-train PLMs on the built tasks (Fig.1b and Methods). In effect, MAML learns to find the optimal initial model parameters such that small changes in them will produce large improvements on the target task.

We have also added two references that utilize MAML to recognize drug-target interactions and antigen binding (Page 14 in the revised manuscript):

Wang et al., ZeroBind: a protein-specific zero-shot predictor with subgraph matching for drug-target interactions, *Nature Communications*, 14, 2023

Gao et al., Pan-Peptide Meta Learning for T-cell receptor–antigen binding recognition, *Nature Machine Intelligence*, 5, 2023

- 9. Results: “MSE denotes fine-tuning the entire PLM with fitness labels as done by Rives et al.”. I thought that Rives et al. simply took the output probability of the amino acids (or rather the difference thereof) for approximating mutation effect, i.e., there was no finetuning involved from what I remember. Maybe double check and clarify/rewrite if needed.**

Response: The work of Rives et al. has a supervised learning task for predicting mutational effect. When finetuning ESM-1b, they regress the (scaled) mutational effect with the likelihood ratio computed by ESM-1b (Page 8 in their Supplementary Information).

- 10. Results: “Meanwhile, LTR + LoRA outperforms the above methods on all training set sizes in terms of both Spearman correlation and NDCG”. Make clear that this is a bit circular: you replace MSE by ranking-loss, benchmark via metrics that solely care about ranking and you do better. This is fine but it should be stated very clearly.**

Response: We have added two metrics that take the actual label values into account: Pearson correlation and MAE (Page 5 in the revised manuscript and Fig.R2c,d in the first reviewer’s comment). FSFP effectively improves the performance of ESM-2 in terms of Pearson correlation but cannot optimize for MAE. Generally, the goal of protein engineering is to identify mutants with enhanced fitness, so ranking-related metrics assume greater importance. On the other hand, in the context of few-shot learning, accurately predicting exact label values becomes challenging due to significant differences in their range between training and testing data. In fact, the regression-based methods exhibit substantial MAE as well (Fig.R2d), rendering their absolute output values also impractical for real-world applications. Since the order matters more, the ranking-related metrics are favored for fitness prediction and thus LTR is more suitable.

11. Fig 2a and 2b show identical trends. Adding panel B only adds minor information. Would consider moving to SOM.

Response: We thank Reviewer2 for this suggestion. Fig.2b has been replaced with the distribution of the performance improvement over zero-shot prediction (see Fig.R1 in the first reviewer’s comment). Original Fig.2b has been moved to SI.

12. Discussion: “For example, when predicting the properties highly correlated to protein structure such as binding and thermostability, one can select SaProt”. Any way to back this up with numbers/examples?

Response: The discussion section has been rewritten and this statement has been removed (Page 12 in the revised manuscript). We have reported the performance of different approaches by the taxon and function of the target protein in Fig.R3c,d in the first reviewer’s comment.

13. Add in main text which ESM version was used. Consider benchmarking other versions of ESM as well. It would be interesting to see the effect of model size on your benchmark, i.e., does model size really constantly lead to better downstream performance?

Response: Thanks for the comment. We have made clear that the 650M version of ESM-2 was used (Page 5 and 8 in the revised manuscript). We have also analyzed the impact of changing the model size (35M, 150M, 650M and 3B) of ESM2 on FSFP, and find that FSFP keeps achieving better performance on larger model (Page 5 in the revised manuscript and Fig.R2a in the first reviewer’s comment).

14. How were structures computed for SaProt?

Response: To build the inputs for SaProt, we obtained the structures of the proteins via AlphaFold2 or download from AlphaFoldDB if available, and this has been described on Page 15 in the manuscript.

15. Given that LORA injects adapters in each layer, the architecture of the underlying pLMs can have an impact on how many trainable parameters each model has at the end. State clearly how many trainable parameters each of the compared models has.

Response: For each of the compared PLMs, the 650M version is chosen for evaluation, where the trainable LoRA parameters account for 1.84% (the difference between the actual numbers are small) of the entire model. This statement has been added to Page 8 in the revised manuscript.

16. Make absolutely clear that protein gym compares methods that never used any DMS data for training while you did. Once you did this, you could also put your method into perspective of fully unsupervised methods compared in the public leaderboard (of course, always with the remark of the difference between unsupervised vs supervised approach).

Response: Thanks for the comment. On page 5 in the revised manuscript, we have stated that “ProteinGym is originally used for evaluating the zero-shot performance of PLMs, and we turn it into a few-shot learning benchmark as follows.”

17. I appreciate that you visualized all your results but consider adding a table with numbers and standard errors to SOM to have exact numbers to compare (might be important for others in the future who would like to compare to your method).

Response: We thank Reviewer2 for this suggestion. The detailed performance of different approaches on each dataset has now been uploaded to the public GitHub repository (Page 17 in the revised manuscript).

18. Did not run the code but the README of the github suggests that reported performances can be reproduced. Given the large numbers of parameters available, I suggest to provide a

script that allows to reproduce the numbers reported in the paper.

Response: We have updated the default hyperparameters of the code and provided a bash script that automatically runs the pipeline of our benchmark. Directly run it according to the guidance should reproduce the results.

REVIEWER COMMENTS

Reviewer #1 (Remarks to the Author):

I have no further comments.

Reviewer #2 (Remarks to the Author):

Thanks a lot for your thorough revision. All my concerns were addressed.

Only some minor things that came up during the changes introduced via the revision:

- it is very interesting to see that your approach yields the exact same mutant as the one proposed by the zero-shot approach (T441L, Table R2). I think this is worth noting this prominently, especially, as you keep arguing throughout the manuscript that the ranking of candidates is more important than their precise values (which I agree but after all if your complicated method reaches at the end of the day the exact same highest-scoring mutant, i.e., the one that's ranked best, then I think this should be stated somewhere. I understand that you increase the total number of mutants that are better than WT but I think equally important is the highest scoring variant that can be found via a method (so good that you already highlight that your method increases the number of good mutants but maybe also mention in the same or next sentence that this did not lead to a mutant that improved overall scoring).

- In the revision you mention that "The underlying reason is that we use the target training data to early stop meta". I think this is an important piece of information. In Figure R3 you then write that " The base model is ESM-2 and the target training set size is 40. ". Does that mean that you computed early stopping just on those 40 proteins? - After all, in general, none of the samples used to determine early stopping should be used for any sort of benchmarking. Given that I have to admit that your method becomes a bit circular and I keep losing overview on what fraction of data you used at what point, I would just ask you to make absolutely sure that all benchmarks adhere to: a) none of the proteins used for early stopping were used for benchmarking within a given cross-validation split and b) if you limit the number of samples of the target protein used to optimize the model, make sure to factor in the number of early stopping samples into this number. After all, this is what would happen in a real-life scenario: you have e.g. 20 measurements and with those 20 you need to do everything, from training to early stopping etc. Reason why I bring this up: I find it really irritating how much your method benefits from training on a completely unrelated proteins with very different functions than what you are trying to optimize (at least, this is my take away from Fig. R3a. - also in Fig. R3a: from what I understood, you chose the protein with lowest sequence similarity as aux task here. Maybe then rather write in the legend sth like "low seq. sim" instead of "low task sim" (after all the two might be related but one you can quantify and the other is rather vague).

Response to Reviewer2

- 1. it is very interesting to see that your approach yields the exact same mutant as the one proposed by the zero-shot approach (T441L, Table R2). I think this is worth noting this prominently, especially, as you keep arguing throughout the manuscript that the ranking of candidates is more important than their precise values (which I agree but after all if your complicated method reaches at the end of the day the exact same highest-scoring mutant, i.e., the one that's ranked best, then I think this should be stated somewhere. I understand that you increase the total number of mutants that are better than WT but I think equally important is the highest scoring variant that can be found via a method (so good that you already highlight that your method increases the number of good mutants but maybe also mention in the same or next sentence that this did not lead to a mutant that improved overall scoring).**

Response: Thanks for the advice. We have stated this clearly on Page 10 in the revised manuscript as below. Specifically, the best mutant (i.e., the one with the highest T_m value) found by ESM-1v (FSFP) is also recommended by ESM-1v (Zero-shot). However, among the positive mutants predicted by ESM-1v (FSFP), 9 of them do not appear in the training data, suggesting that FSFP can enable PLMs to identify more protein variants that are better than wild type.

- 2. In the revision you mention that "The underlying reason is that we use the target training data to early stop meta". I think this is an important piece of information. In Figure R3 you then write that " The base model is ESM-2 and the target training set size is 40. ". Does that mean that you computed early stopping just on those 40 proteins?**

Response: Yes. The meta-training process (training on the data from the auxiliary tasks) is stopped according to the validation performance on the 40 training examples from the target protein. These 40 training examples are then used to keep fine-tuning the meta-learned model. The fine-tuning process is also stopped based on these 40 examples. Please see the reply below for details.

- 3. After all, in general, none of the samples used to determine early stopping should be used for any sort of benchmarking. Given that I have to admit that your method becomes a bit circular and I keep losing overview on what fraction of data you used at what point, I would just ask you to make absolutely sure that all benchmarks adhere to: a) none of the proteins used for early stopping were used for benchmarking within a given cross-validation split and b) if you limit the number of samples of the target protein used to optimize the model, make sure to factor in the number of early stopping samples into this number.**

Response: a) During the transfer learning process (training on the data from the target protein), the examples used for early stopping are *split from the target training data* but not from the testing data (where we compute the performance metrics), and there is no data leakage in our benchmark. b) We achieve early stopping by using the training set itself and do not require an extra validation set. Given a training set, we first adopt a cross-validation scheme on this dataset to estimate the number of training iterations, and then train that number of iterations on the whole data of it. This procedure has been detailed in the Method section (Page 16 in the manuscript), and also copied below.

When sufficient labeled data are available, early stopping is generally based on a separate validation set. However, a held-out validation set may result in insufficient training data in a low-resource scenario. On the other hand, if the validation data size is assigned too small, the validation scores such as Spearman correlation may not be representative enough for early stopping. Based on these considerations, we propose to estimate the number of training iterations for transfer learning by Monte Carlo cross-validation. Specifically, we create 5 random splits of *the training set* into training and validating data. The proportion of training and validating data is 0.5:0.5 when the training data size is less than

50 otherwise 0.75:0.25. For each split, the model is trained on the sub-sampled training data for up to 500 steps, and we record the Spearman correlation calculated on the validating data every 5 steps. After 5 rounds of training and validating, we choose the training step number with the highest average validation score across different splits, and finally train the model for that number of steps on the whole training data.

The training data from the target protein is also used to early stop the meta-training procedure. Similarly, 5 random splits of it are generated first. For every 5 steps of the outer optimization during meta-training, we train the current meta-learned model $f_{\omega, \theta}$ on the sub-sampled target training sets for 5 gradient steps (same to the inner optimization) and compute the validation Spearman scores. We stop meta-training if the average validation score of different splits does not improve within 20 consecutive records and pick the best meta-learner according to this score.

4. also in Fig. R3a: from what I understood, you chose the protein with lowest sequence similarity as aux task here. Maybe then rather write in the legend sth like "low seq. sim" instead of "low task sim" (after all the two might be related but one you can quantify and the other is rather vague).

Response: Thanks for the suggestion. The legend of Fig.S7a have been changed to “FSFP (low protein similarity)”. Since the embeddings computed by PLMs can also have structural information, we assume that “protein similarity” would be better than “sequence similarity”.

REVIEWERS' COMMENTS

Reviewer #2 (Remarks to the Author):

Thanks for the swift reply; all my comments were addressed.